# Electrophysiological markers for anticipatory processing of nocebo-augmented pain

**Joseph S. Blythe**[1,2]*, **Kaya J. Peerdeman**[1,2], **Dieuwke S. Veldhuijzen**[1,2], **Julian D. Karch**[3], **Andrea W. M. Evers**[1,2,4,5,6]

1 Health, Medical & Neuropsychology Unit, Institute of Psychology, Leiden University, Leiden, The Netherlands, 2 Leiden Institute for Brain and Cognition, Leiden, The Netherlands, 3 Methodology and Statistics Unit, Institute of Psychology, Leiden University, Leiden, The Netherlands, 4 Department of Psychiatry, Leiden University Medical Centre, Leiden, The Netherlands, 5 Medical Delta Healthy Society, Delft, The Netherlands, 6 Leiden University, Technical University Delft, & Erasmus University Rotterdam, Delft, The Netherlands

* j.s.blythe@fsw.leidenuniv.nl

**Data Availability Statement:** All data from the study are available in the BioStudies database (accession number S-BSST1144).

**Funding:** This work was supported by a Dutch Research Council (NWO) Vici grant (#45316004)

## Abstract

Nocebo effects on pain are widely thought to be driven by negative expectations. This suggests that anticipatory processing, or some other form of top-down cognitive activity prior to the experience of pain, takes place to form sensory-augmenting expectations. However, little is known about the neural markers of anticipatory processing for nocebo effects. In this event-related potential study on healthy participants (n = 42), we tested whether anticipatory processing for classically conditioned nocebo-augmented pain differed from pain without nocebo augmentation using stimulus preceding negativity (SPN), and Granger Causality (GC). SPN is a slow-wave ERP component thought to measure top-down processing, and GC is a multivariate time series analysis used to measure functional connectivity between brain regions. Fear of pain was assessed with the Fear of Pain Questionnaire-III and tested for correlation with SPN and GC metrics. We found evidence that both anticipatory processing measured with SPN and functional connectivity from frontal to temporoparietal brain regions measured with GC were increased for nocebo pain stimuli relative to control pain stimuli. Other GC node pairs did not yield significant effects, and a lag in the timing of nocebo pain stimuli limited interpretation of the results. No correlations with trait fear of pain measured after the conditioning procedure were detected, indicating that while differences in neural activity could be detected between the anticipation of nocebo and control pain trials, they likely were not related to fear. These results highlight the role that top-down processes play in augmenting sensory perception based on negative expectations before sensation occurs.

## Introduction

The human brain's capacity to anticipate future events is an adaptive function that allows information from previous experience to aid in predicting future outcomes [1]. Particularly for threatening stimuli in the environment, such as those associated with pain or injury, the

awarded to A.W.M. Evers. Website: nwo.nl. The funders had no role in study design, data collection and analysis, decision to publish, or preparation of the manuscript.

**Competing interests:** The authors have declared that no competing interests exist.

brain's ability to detect, anticipate, and prepare for such encounters is well established [2]. Nocebo effects on pain may be thought of as a negative consequence of this function, in which the anticipation of a painful experience, heightened by prior painful experiences or other learning, exacerbates future pain experiences [3, 4]. Nocebo effects are thought to worsen pain symptoms in clinical conditions and everyday life, and a better understanding of how anticipatory processing substantiates these effects would aid in their management.

Nocebo effects on pain are thought to arise from a learned expectation that a given stimulus, e.g., a medical procedure, causes an increased pain response [5]. In laboratory settings, this learning is typically modeled through paradigms including classical conditioning, verbal suggestion, and observational learning, or a combination of these [6]. Classical conditioning with verbal suggestion, the most commonly used nocebo paradigm, employs a combination of instructional and associative learning to present and reinforce an association between a formerly neutral stimulus (e.g. activation of a sham medical device) and an increased pain response [7–9].

Efforts to study anticipatory processing for threatening and aversive stimuli have identified components in event related potentials (ERPs) which are thought to signify anticipatory neural activity [10–12]. One component, stimulus preceding negativity (SPN), can be measured several seconds prior to the presentation of an anticipated stimulus, and is considered to be an indicator of mental processing related to the anticipated stimulus [13, 14]. SPN presents as a slow negative component, and its amplitude has been shown to increase with greater intensity or uncertainty of the anticipated pain stimulus [15]. Late SPN amplitudes, measured 500ms prior to a pain stimulus, are decreased when receiving a placebo treatment compared to a no-treatment control condition [16]. If late SPN amplitudes can also discriminate between the anticipation of nocebo and control pain stimuli, this would demonstrate top-down cognitive influence for nocebo-augmented pain perception, and provide a neural marker for this process.

While SPN is maximally expressed at electrode Cz prior to the presentation of an anticipated stimulus [10], anticipatory processing may also occur as a distributed process across numerous regions of the brain. Expectations are thought to exert top-down control over bottom-up processes like sensory perception [17, 18], influencing the order and depth of processing for incoming sensory precepts. Communication between frontal, executive regions and temporoparietal regions encompassing the somatosensory cortex would substantiate this process [3, 19]. Granger Causality, a measure of how past events in time series X may predict future events in time series Y, offers a method of quantifying functional connectivity, or the transfer of information between regions in the brain [20–22]. This method has previously been applied to classify the electrophysiological signature of different pain intensities [23], and differences in pain processing between individuals with and without chronic migraines [24], but not to study nocebo, placebo, or other expectancy effects related to pain. By measuring functional connectivity during the experience of nocebo effects on pain, we can build on previous work that identified regions of interest for nocebo effects, and observe how these regions interact to shape pain sensations. Measuring both SPN and Granger Causality allows us to extend a validated measure of anticipatory processing to nocebo-augmented pain, and explore whether new measures like Granger Causality can shed further light on how this processing occurs throughout the brain. These tools could help us assess how cognitive processes are exacerbating pain and other symptoms, and whether psychological interventions for such symptoms are having an impact on these top-down processes.

On top of learned associations, affective states such as fear can influence the anticipation of pain and nocebo responses. Fear of pain may exacerbate nocebo effects through increased anticipatory processing of the fear-inducing pain stimulus, similar to the effect of fear on

anticipatory processing in social phobia [25, 26]. By focusing cognitive resources on a nocebo-associated stimulus, fear may spur stronger associations between the stimulus and increased pain responses. However, findings for a relationship between fear of pain and the magnitude of the nocebo effect are mixed [27–29], despite the well-documented impacts of fear of pain in other contexts [30].

In this study, we investigated the role of anticipatory processing in nocebo effects on pain, and its relation to fear of pain. Using a classical conditioning with verbal suggestion paradigm, we induced nocebo effects on thermal pain while recording electroencephalography (EEG), allowing us to investigate the role of anticipatory processing through measures of SPN and Granger Causality. We hypothesized that after acquisition of a nocebo effect, we would observe larger late SPN amplitudes for nocebo stimuli, relative to control, measured 500ms prior to the pain stimulus. Additionally, we hypothesized stronger GC values in frontal to temporoparietal connections for nocebo stimuli relative to control. We also hypothesized that fear of pain scores would correlate with self-reported magnitude of the nocebo effect, and that both would correlate with late SPN amplitudes, and GC values.

## Method

### Ethics

The study was approved by the Leiden University Psychology Research Ethics Committee (2021-11-03-A.W.M. Evers-V1-3520), and conducted in accordance with the Declaration of Helsinki. The hypotheses were preregistered in the Netherlands Trial Registry: https://trialsearch.who.int/Trial2.aspx?TrialID=NL9813.

### Participants

A sample of healthy adult volunteers was recruited via advertisements in university buildings and online (SONA Systems, Tallin, Estonia) from April-December 2021. English speaking individuals aged 18–35 with (corrected to) normal vision, no current analgesic- or psychotropic medication use, and no current or chronic pain symptoms or psychiatric illness were eligible to participate. Individuals at high risk for severe COVID-19, with recent COVID-19 exposure, and pregnant women were ineligible. Participants were instructed not to consume more than three units of alcohol or any recreational drugs 24 hours prior to the experiment. The inability to rate pain stimuli of the same temperature within +/- 2 points of the same rating during calibrations (e.g. first rating a 50˚C stimulus as a 7 and then rating it as a 4), led to exclusion due to inconsistent pain perception, as did failing to rate any of the pain stimuli as at least 6 on a 0–10 pain intensity scale.

The sample size was determined by a power analysis based on a study measuring SPN amplitudes for placebo effects on pain [16]. In a within-subjects design, Morton et al. observed a 3.3μV mean difference in late SPN between placebo and control pain trials, $d$ = .47. With $\alpha$ = .05 and 1-$\beta$ = .80, a sample of 38 participants was needed to reliably test for this effect. Based on anticipated dropout and unusable data (e.g. from artefacts) we selected a sample size of 42 participants.

### Materials

**Pain stimuli.** Thermal pain was induced on the non-dominant volar forearm with a contact heat evoked potential stimulator (CHEPS) 27mm diameter thermode connected to a PATHWAY device (Medoc Advanced Medical Systems, Ramat Yishai, Israel). The CHEPS thermode ramps up from a baseline temperature (32˚C) to a target temperature at 70˚C/

second, remains at target for 300ms, and returns to baseline at 40°C/second. Using previously validated methods [28], warmth and pain perception threshold temperatures, and the temperatures needed to induce distinct moderate and high pain intensities as the control and nocebo pain stimuli, respectively, were determined. Participants rated a series of thermal stimuli on a 0–10 pain intensity scale, with 0 indicating no pain at all and 10 indicating the worst imaginable pain on their arm, to assess what temperatures were reliably rated as moderate and highly painful. Moderate pain stimuli were approximately 4 on the 0–10 scale, while high pain stimuli were approximately 7.

**Nocebo manipulation.** Participants were led to believe that a commercially available transcutaneous electric nerve stimulator (TENS; Beurer EM80, Ulm, Germany) would increase their sensitivity to thermal pain. This was achieved with a classical conditioning and verbal suggestion paradigm. For the verbal suggestion, participants were told that a purple square displayed on a computer monitor with the message "ON" would indicate the activation of the TENS device during the following pain stimulus, which would increase their sensitivity to pain. They were also told that a grey square displayed on the screen with the message "OFF" indicated that the TENS device was not active and their pain sensitivity would not increase. Two TENS electrodes were placed on the participant's non-dominant volar wrist and upper volar forearm, and the participant felt several pulses from the TENS device during a mock calibration. The device was surreptitiously deactivated for the remainder of the experiment.

The classical conditioning component of the nocebo manipulation included an acquisition phase with 80 pain trials, and an evocation phase with 62 pain trials, similar to prior experiments [6, 7, 31–34]. During acquisition, 40 control trials and 40 nocebo trials were presented in pseudorandom order, with no more than three consecutive trials of either type. Control trials always paired a grey square and the text "OFF" with a moderate pain stimulus, and nocebo trials always paired a purple square and the text "ON" with a high pain stimulus. During evocation, 30 control trials and 30 nocebo trials were presented in pseudorandom order, and each was now paired with only the moderate pain stimulus. Two additional nocebo trials paired with the high pain stimulus, reinstatement trials, presented approximately halfway through the evocation phase were included to prolong the nocebo effect. For all trials, the grey or colored square was displayed onscreen for 5000ms. To signal the approaching pain stimulus, a short high-pitched tone (1000Hz, 250ms duration) played from speakers in front of the participant three times, -3500ms, -2500ms, and -1500ms prior to the pain stimulus onset. Following each pain stimulus, participants gave a 0–10 pain intensity rating via a keyboard. Between each trial, a fixation cross was displayed for a jittered duration of 5000ms +/- 2000ms.

**EEG.** EEG was recorded with a BioSemi ActiveTwo electrode system (Biosemi, Inc., Amsterdam, the Netherlands) with 32 scalp electrodes in a BioSemi head cap, arranged in the international 10–20 system. Data were acquired with a sampling rate of 1024Hz, and bandpassed filtered online from 0.1 to 100Hz (with 100Hz low-pass and 0.01Hz high-pass hardware filters).

## Measures

**Pain intensity.** Pain intensity ratings were collected with a 0–10 numeric rating scale (NRS) presented on a computer screen. Zero was labeled as "no pain at all" and 10 as "worst pain imaginable on my arm". Ratings could be given in increments of 0.5, and were given via buttons on a keyboard. Pain ratings were averaged across trials and participants by trial type, resulting in mean values for control and nocebo trials, split between acquisition and evocation phases.

**SPN.** Stimulus preceding negativity was measured from electrode Cz in a 500ms window directly preceding the pain stimulus, in line with previous work [16]. The signal, expressed

in µV, from Cz during this window was averaged across trials and participants by trial type, resulting in mean values for control and nocebo trials, split between acquisition and evocation phases.

**Granger Causality.** Granger Causality (GC) was used to measure functional connectivity and information transfer -1000ms to +1000ms relative to the onset of the pain stimulus. Using a vector autoregressive model, GC tests whether past information in time series X can predict later information in time series Y better than past information in time series Y alone. Based on earlier work that indicates descending pain modulation starts in frontal regions and descends throughout other pain processing regions, including bilateral insula and somatosensory cortex [35–38], we tested an a priori network with four nodes: frontal left, frontal right, temporoparietal left, temporoparietal right. The mean GC indices of left and right frontal to left and right temporoparietal nodes, as well as time-series of GC indices across the 2s window, were compared between nocebo and control trials, separately for acquisition and evocation phases.

**Fear of pain.** Fear of pain was measured with the Fear of Pain Questionnaire-III [39]. The questionnaire consists of 30 items describing painful experiences which are rated on a 1–5 Likert scale with 1 indicating "no fear" and 5 indicating "extreme fear". FPQ-III scores were calculated by summing the scores of all 30 items. Scores range from 30–150 with higher scores indicating a greater fear of pain.

**Manipulation checks.** Two manipulation check questions were asked to assess the degree to which participants believed the experimental manipulations. On a 0–100 scale with 0 indicating 'completely disagree' and 100 indicating 'completely agree', participants rated their agreement with the statements 'I trusted the experimenters' and 'at the start of the experiment, I noticed that the TENS device made my pain worse'.

## Procedure

Potential participants registered for the experiment via SONA (SONA Systems, Tallinn, Estonia). An experimenter sent potential participants an online screening checklist (Qualtrics, Provo, Utah, USA) to ensure eligibility to participate. Eligible participants were invited to the lab for a single, two-hour session.

Upon arriving at the lab, participants gave written informed consent to take part in the experiment. Participants' age and gender data was collected, followed by measurement of warmth and pain thresholds, and calibration of the moderate and high pain stimuli. The EEG was then set up, followed by the mock calibration of the TENS device, and the verbal suggestion for the increased pain sensitivity it would cause when activated. Participants completed the classical conditioning paradigm, lasting approximately 45 minutes. All equipment was then removed from the participant, who completed the FPQ-III and the two manipulation check questions. The participant was debriefed and compensated for their time with cash or course credits according to standard rates set by the university.

## Analysis

Pain ratings, demographics, survey, and manipulation check data were analyzed with SPSS (version 25, Armonk, NY). EEG data was preprocessed in MATLAB (version 2021A) (Natick, MA), and analyses were run in MATLAB and R (version 4.1.3; [40]). P values less than .05 were considered significant except in the case of GC analyses, where corrections for multiple tests were implemented (described later). Histograms for pain ratings by trial type and FPQ-III scores were visually inspected for normality and outliers. Outliers were confirmed with *Z* scores. Normality was tested with Shapiro-Wilks tests. Granger Causality estimates were checked for symmetrical distribution.

**Nocebo effect.** Nocebo effects were measured as the difference between all nocebo and all averaged control trials during the evocation phase, excluding the reinstatement trials. A paired sample *t*-test was used to compare the mean pain rating of nocebo trials to the mean pain rating of control trials.

**SPN.** Preprocessing and analysis of EEG data pertaining to SPN magnitudes was conducted with EEGLAB [41] and ERPLAB [42]. In EEGLAB, recordings were downsampled to 256Hz and visually inspected for artifacts in channel Cz. Data were rereferenced to the average signal and independent components analysis was used to identify eye and muscle artifacts. EEG recordings were then segmented into epochs of 6000ms, beginning 5000ms before the onset of the pain stimulus and ending 1000ms after onset for all pain stimuli, excluding epochs for the two reinstatement trials. Epochs were baseline corrected from -1000ms to the onset of the colored square denoting control or nocebo trial type. The grand average ERP was computed by trial type (control or nocebo, separately for acquisition and evocation phases) and conditioning phase (acquisition or evocation), and paired samples *t*-tests were used to compare control and nocebo SPN amplitudes for acquisition and evocation phases. The difference in control and nocebo grand averages was tested for correlation with self-reported pain ratings with a Pearson's correlation.

**Granger Causality.** Preprocessing of EEG data pertaining to Granger Causality analyses was conducted with EEGLAB and ERPLAB, separately from preprocessing for SPN. In EEGLAB, recordings were downsampled to 512Hz, visually inspected for artifacts and bad channels, and rereferenced to the average signal. Independent components analysis was used to identify and remove eye and muscle components from the recordings. EEG recordings were segmented into epochs starting 1000ms before the onset of the pain stimulus to 1000ms afterwards, and baseline corrected from -1500ms to -1000ms relative to the start of the pain stimulus. In ERPLAB, data from electrodes Fp1, AF3, F3, F7 were averaged to form the frontal left node, data from Fp2, AF4, F4, F8 were averaged to form the frontal right node, data from T7, P7, CP1 CP5 were averaged to form the temporoparietal left node, and data from T8, P8, CP2, CP6 were averaged to form the temporoparietal right node.

These data were then analyzed with the MVGC toolbox [43] in MATLAB. Due to potential nonstationarity in ERP data, a vertical regression model was used [44]. The data now consist of 1024 samples of EEG signal (2000ms of data at 512 Hz) for each of the four nodes in the model. Using a window of length 50ms, slid across the data in steps of 10ms, 98 GC values were computed for each pair of nodes. To determine whether GC values indicated significant information transfer between nodes, GC values were compared against a significance threshold set with a Šidák correction [45]. GC values were computed separately per participant, then means across the sample and time series by trial type were computed. Finally, mean GC values were compared between trial types (nocebo and control) separately for acquisition and evocation phases with Wilcoxon sign rank tests. GC time series across the 2000ms epoch were compared on differences between nocebo and control GC time series with an adaptable regularized Hoteling's $T^2$ (ARHT) test [46], from the ARHT package (https://CRAN.R-project.org/package=ARHT) in R (R Core Team, 2022). The ARHT test is an adaptation of Hotelling's *t* test that is suited for data with dimensionality greater than the sample size.

**Fear of pain.** Pearson's correlations were used to test for relationships between FPQ-III scores and self-reported nocebo effects, SPN mean amplitudes and mean GC values.

# Results

## Participants

A sample of 65 participants was invited to the lab, of which 23 were excluded during pain calibrations (15 due to inconsistent pain ratings during calibrations and 8 who did not rate any

calibration stimuli at least a 6 on a 0–10 NRS of pain intensity), resulting in 42 participants who completed the experiment. All participants attested that they did not consume more than three alcoholic beverages or other recreational drugs 24 hours prior to their lab session. EEG data from 2 participants was not usable due to excessive movement artefacts, and their data were excluded from all analyses. The final sample ($N$ = 40, $M_{age}$ = 22.3, $SD$ = 3.8) consisted of 26 (65%) female, 13 (32.5%) male, and 1 (2.5%) nonbinary participants. The mean warmth perception threshold was 35.3˚C ($SD$ = 1.2), and the mean pain perception threshold was 42.4˚C ($SD$ = 2.3). The mean moderate temperature, used for control trials in the acquisition phase and both control and nocebo trials in the evocation phase, was 48.1˚C ($SD$ = 1.1). The mean high temperature, used for nocebo trials during the acquisition phase, and the two reinstatement trials during evocation, was 50.1˚C ($SD$ = 1.1). Regarding belief in the manipulation, participants generally reported that the TENS device made their pain worse (0–100 scale completely disagree to completely agree, $M$ = 70.8, $SD$ = 27.8). Participants generally trusted the experimenter (0–100 scale no trust in experimenter to complete trust in experimenter, $M$ = 81.9, $SD$ = 23.2).

## Nocebo effect

First, we tested for the presence of a nocebo effect during the evocation phase. No significant derivation of normality was found for nocebo and control trial pain ratings (all $p$ > .1), nor were influential outliers observed. Mean pain ratings for nocebo trials ($M$ = 5.45, $SD$ = 1.53) were significantly higher than for control trials ($M$ = 4.86, $SD$ = 1.47), across the entire evocation phase [$t(39)$ = 4.12, $p$< .001, $d$ = .90], as depicted in Fig 1.

## SPN

SPN amplitudes were not normally distributed for control trials in the evocation phase ($p$ = .022), as well as nocebo ($p$ = .001) and control ($p$ = .018), trials in the acquisition phase, due to the presence of a single outlier detected when assessing normality ($Z > 3$ for control and nocebo acquisition, $Z$ = 2.65 for evocation control). Removing the outlier restored normality but did not change the significance of the results, and because the nocebo-control difference scores were normally distributed (all $p$> .2), the outlier was included. Upon visual inspection of the data, it was considered suboptimal to measure SPN only in the -500 to 0ms window relative to the pain stimulus onset, as a waveform from an auditory cue was present for control trials but not nocebo trials during within this window (Fig 2). While the timing of events for nocebo evocation trials was correct, for the control trials there was about 400ms less time between the final auditory cue and the pain stimulus due to a technical issue. SPN was instead measured for the entire 4s anticipatory period so that for each trial type all waveforms for the auditory cues were included equally, partially addressing this fault. In this window, SPN was greater for nocebo trials ($M$ = -1.61μV, $SD$ = 3.41) relative to control trials ($M$ = -0.25 μV, $SD$ = 3.99) during the evocation phase at electrode Cz, ([$t(39)$ = 3.22, $p$ = .003, $d$ = .51], Fig 2). When using the a priori 500ms window, we see a similar result ($M_{nocebo}$ = -1.03 μV, $SD_{nocebo}$ = 6.54, $M_{control}$ = 0.42 μV, $SD_{control}$ = 6.31, [$t(39)$ = 2.24, $p$ = .031, $d$ = .35], This effect was also observed during the acquisition phase, where no trial timing issues were present and SPN was measured from -500 to 0ms relative to the pain stimulus trigger ($M_{nocebo}$ = -0.92 μV, $SD_{nocebo}$ = 6.31, $M_{control}$ = 0.51 μV, $SD_{control}$ = 6.21, [$t(39)$ = 2.92, $p$ = .011, $d$ = .42], Fig 3). There was no significant correlation between the difference in SPN amplitudes for nocebo and control trials, and the self-reported difference in pain ratings between trial types (i.e., nocebo effect on pain; $r$ = .05, $p$ = .782).

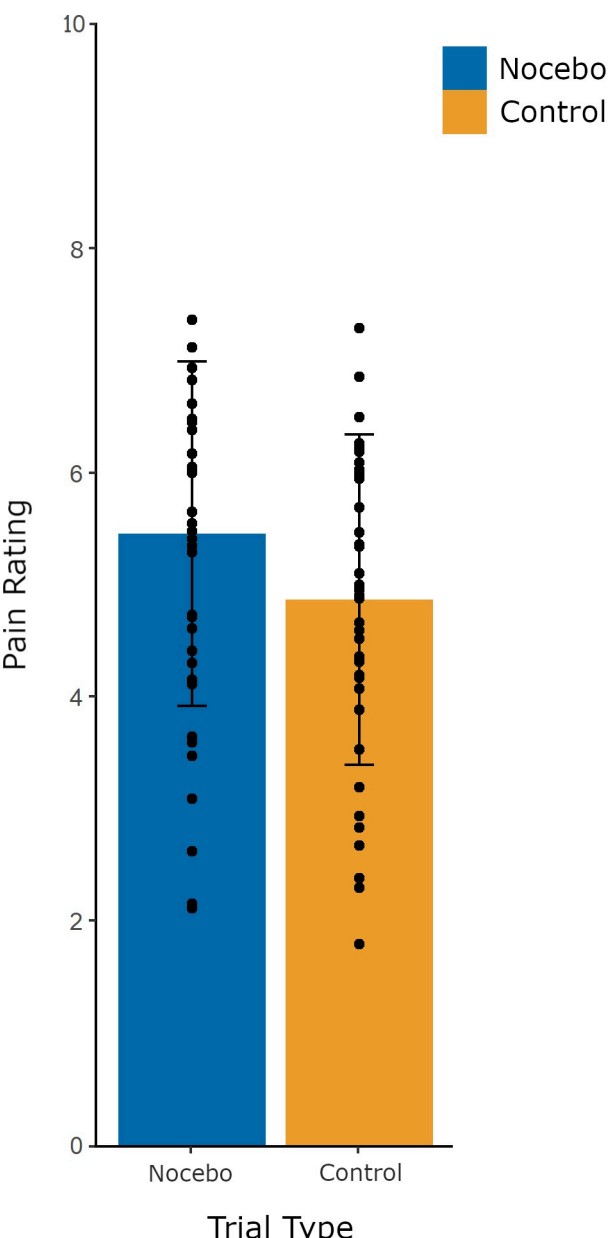

**Fig 1. Mean pain ratings for nocebo and control evocation trials.** Pain ratings were made on a computerized 0–10 NRS. Error bars represent standard deviations. Dots indicate individual participant data.

## Granger Causality

Significance thresholds for Granger Causality estimates within subjects were determined with a Šidák correction separately for each of the 98 windows. The median number of significant windows per node-pair by trial type and across subjects ranged from 27 to 93 (Table 1). Symmetrical distributions were checked for Granger Causality estimates by trial type (nocebo and control) and experimental phase (evocation and acquisition). Wilcoxon sign rank tests found no differences between mean GC estimates between nocebo and control trials in the evocation phase for all pairs of nodes (Table 2). Similarly, no significant differences were detected between nocebo and control trials in the acquisition phase (Table 3).

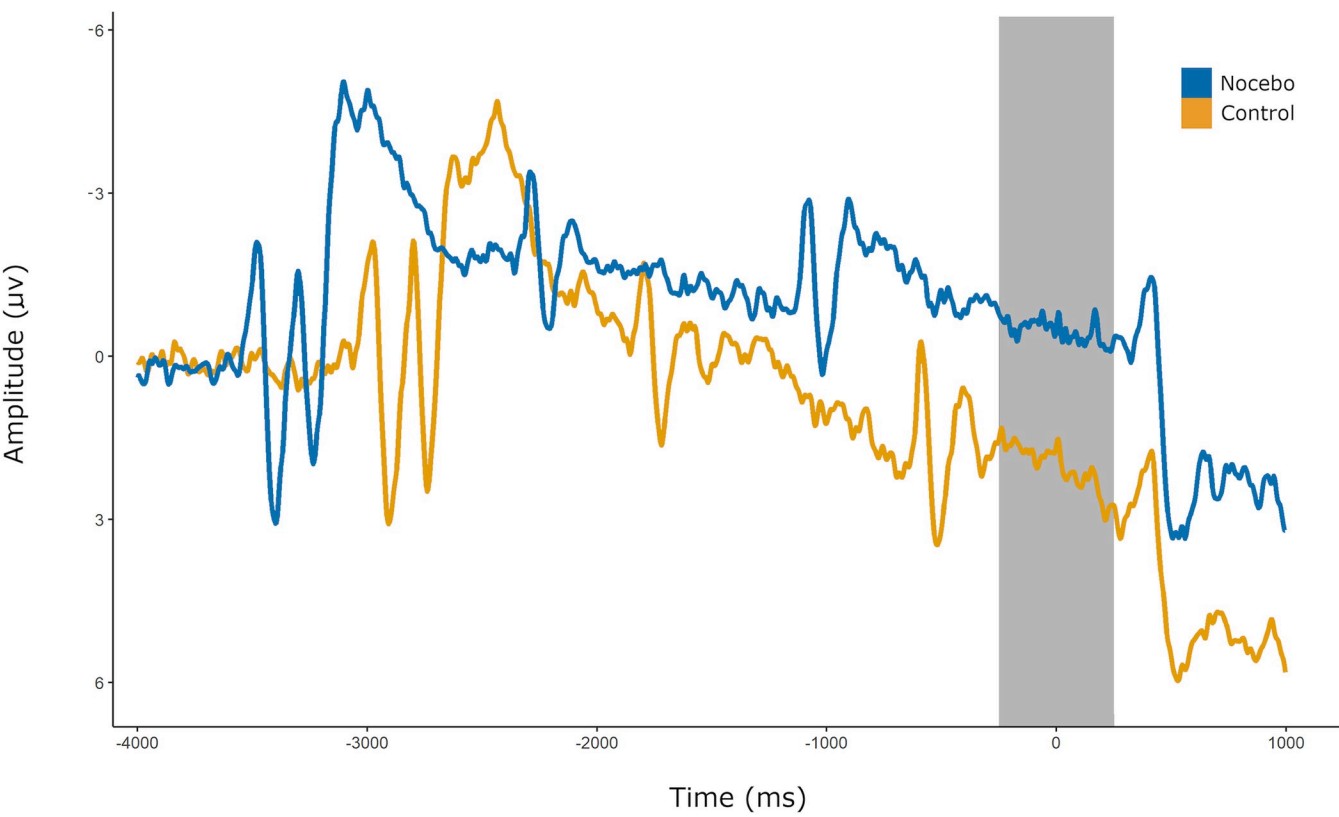

**Fig 2. Mean SPN at Cz during nocebo and control evocation trials.** Mean amplitudes for SPN at electrode Cz for nocebo and control evocation trials. The gray bar indicates the adjusted late SPN window, -500 to 0ms relative to the pain stimulus onset (time 0). Auditory cues were to be presented at -3500, -2500, and -1500ms.

To test for possible differences across the GC time series between control and nocebo trials, we conducted an adaptable regularized Hoteling's $T^2$ (ARHT) test, which detected a difference between the two trial types across time series during the evocation phase ($p = .029$). Visual inspection of the time series (Fig 4) suggests the effect is driven by greater GC estimates for nocebo trials in the frontal right to temporoparietal right path. When applied to the acquisition phase, the ARHT test did not detect a difference across time series between control and nocebo acquisition trials ($p = .70$, Fig 5). We detected no evidence that the trial timing issue described previously impacted these analyses.

## Fear of pain

The mean score on the FPQ-III was 83.1 ($SD = 13.7$). Internal consistency for FPQ-III responses was high, with a Cronbach's $\alpha$ of .88. Scores on the FPQ-III did not correlate with magnitudes of self-reported nocebo effects ($r = .06$, $p = .70$). There were also no correlations between FPQ-III scores and differences in SPN amplitude between nocebo and control trials ($r = .04$, $p = .78$), or differences in GC estimates between nocebo and control trials during evocation (Frontal L to Temporoparietal L: $r = .19$, $p = .248$, Frontal L to Temporoparietal R: $r = .15$, $p = .364$, Frontal R to Temporoparietal L: $r = .14$, $p = .405$, Frontal R to Temporoparietal R: $r = -.06$, $p = .724$).

## Discussion

We investigated the characterization of anticipatory neural processing for nocebo effects on pain, measured with SPN, a slow wave ERP component, and Granger Causality (GC), a

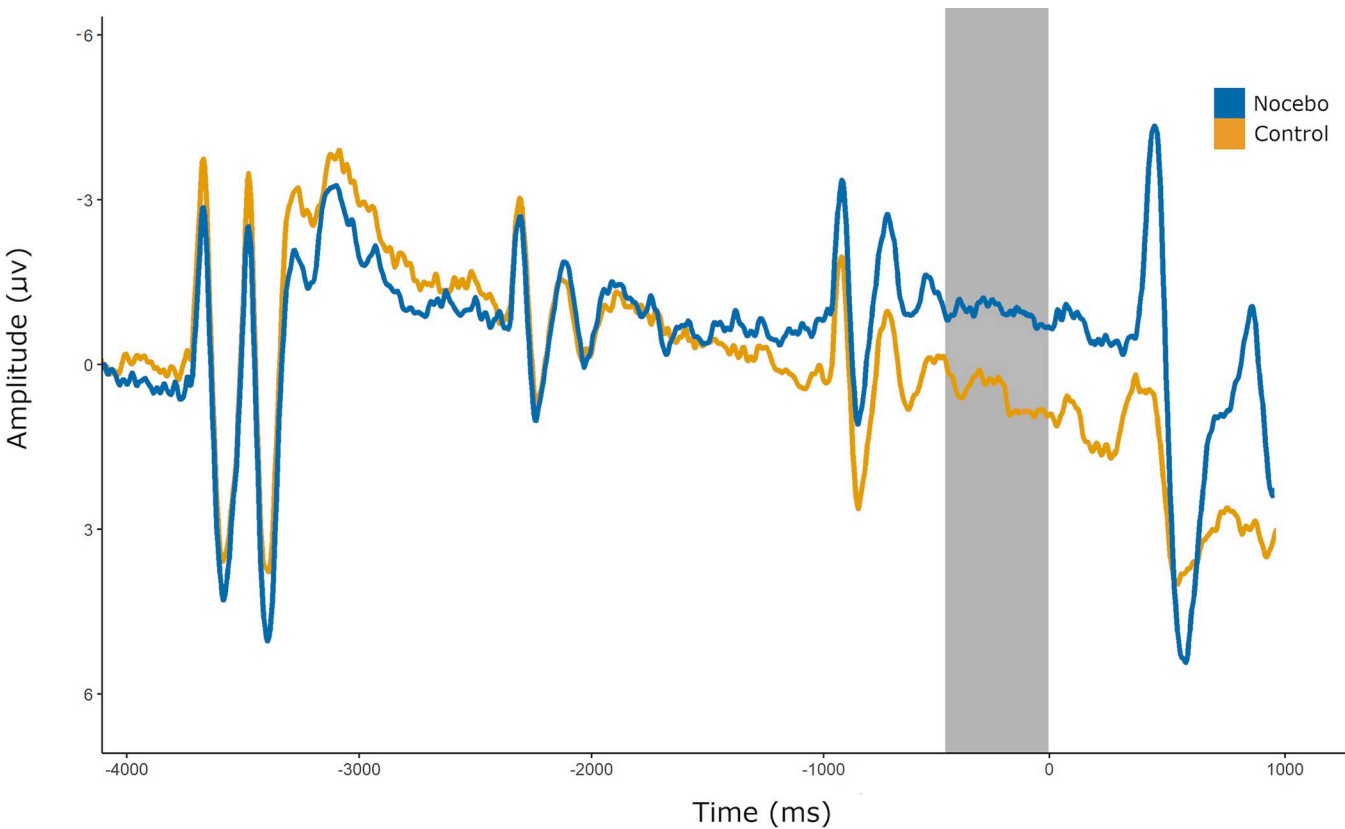

**Fig 3. Mean SPN at Cz during nocebo and control acquisition trials.** Mean amplitudes for SPN at electrode Cz for nocebo and control acquisition trials. The gray bar indicates the late SPN window, -500 to 0ms relative to the pain stimulus onset. Auditory cues were presented at -3500, -2500, and -1500ms, but for the nocebo trials they were erroneously presented 400ms earlier.

measure of information transfer. We also explored the potential link between these measures with fear of pain. During the evocation of nocebo effects, SPN amplitudes were larger in the presence of nocebo-associated stimuli than control stimuli, and we found indications that GC time series varied between trial types, but not mean GC values. During acquisition, the same pattern was observed for SPN, but not for GC. No correlations between SPN amplitudes or GC estimates were detected with nocebo effect magnitudes, measured with self-reported pain intensity. Fear of pain scores did not correlate with SPN amplitudes, GC estimates, or nocebo effect magnitudes.

Although nocebo and control evocation trials are not precisely comparable due to a 400ms difference in the delivery of pain stimuli between trial types, the data suggest that SPN was

**Table 1. Median significant GC windows by trial type.**

| Node pair | Acquisition Control | Acquisition Nocebo | Evocation Control | Evocation Nocebo |
|---|---|---|---|---|
| **Frontal L to Temporoparietal L** | 90.5 | 92 | 27 | 63 |
| **Frontal L to Temporoparietal R** | 91 | 84 | 76 | 85.5 |
| **Frontal R to Temporoparietal L** | 93 | 93 | 58 | 84 |
| **Frontal R to Temporoparietal R** | 61.5 | 65 | 60.5 | 78 |

Results for the median number of significant GC windows by trial type and phase of experiment, out of 98 total windows.

**Table 2. Granger Causality during evocation phase.**

| Node pair | $M_{rank}$ Control | $M_{rank}$ Nocebo | Z score | p |
|---|---|---|---|---|
| Frontal L to Temporoparietal L | 16.61 | 22.10 | -1.04 | .300 |
| Frontal L to Temporoparietal R | 20.18 | 18.95 | -.399 | .690 |
| Frontal R to Temporoparietal L | 17.86 | 21.75 | -.326 | .744 |
| Frontal R to Temporoparietal R | 18.00 | 20.48 | -1.46 | .145 |

Results of comparing mean GC estimates for the evocation phase, spanning -1000ms to +1000ms relative to the onset of the pain stimulus.

larger during nocebo trials relative to control. This is supported by observing the same pattern in the acquisition phase, where the auditory cue timing was matched between trial types. Previously, SPN was also shown to be larger in the presence of placebo stimuli relative to control [16]. Considering the lack of correlation between SPN amplitudes and trait fear of pain, this suggests that more negative SPN amplitudes observed for nocebo stimuli are not driven by fear of pain, and instead measure some other aspect of increased anticipation that is common between placebo and nocebo effects relative to neutral control trials. A future study directly comparing SPN for both placebo and nocebo effects could investigate this possibility, and a study employing an acute measure of state fear instead of trait fear could shed light on any potential link between fear of pain and SPN.

While Granger Causality has previously been used as a means of characterizing different intensities of pain [23], and differences in pain processing between individuals prone to migraines and those not [24], this was its first use for investigating the anticipation of pain, and how that process may change when pain is shaped by a nocebo effect. We had hypothesized that expectations regarding the upcoming pain stimulus would be communicated from frontal to temporoparietal regions, where integration with the bottom-up sensory inputs would shape the pain experience [3, 18, 47]. While significant Granger Causality was detected during the evocation phase for both nocebo and control trials across all four pairs of frontal to temporoparietal nodes (Table 1), no differences between nocebo and control mean GC estimates emerged. For both nocebo and control trials, participants may have held expectations forthe impending pain sensation that coincided with information transfer from frontal to temporoparietal brain regions, which could not be disentangled with this measure. Similarly, no differences in GC estimates between nocebo and control trials were detected during the acquisition phase, when the nocebo effect was actively reinforced with more painful stimuli during nocebo trials. When comparing time-series instead of means, there appeared to be a difference between nocebo and control evocation trials, possibly driven by larger GC estimates for nocebo trials from the frontal right to temporoparietal right nodes, lending some support to our hypothesized increased information transfer from frontal to temporoparietal regions for nocebo over control trials. This could indicate a top-down process by which negative expectations activate in higher order cognitive regions and signal down to sensory processing regions

**Table 3. Granger Causality during acquisition phase.**

| Node pair | $M_{rank}$ Control | $M_{rank}$ Nocebo | Z | p |
|---|---|---|---|---|
| Frontal L to Temporoparietal L | 21.05 | 17.38 | -1.34 | .180 |
| Frontal L to Temporoparietal R | 20.05 | 18.75 | -1.02 | .307 |
| Frontal R to Temporoparietal L | 21.65 | 16.20 | -1.85 | .064 |
| Frontal R to Temporoparietal R | 20.56 | 18.55 | -.01 | .994 |

Results of comparing mean GC estimates for the acquisition phase, spanning -1000ms to +1000ms relative to the onset of the pain stimulus.

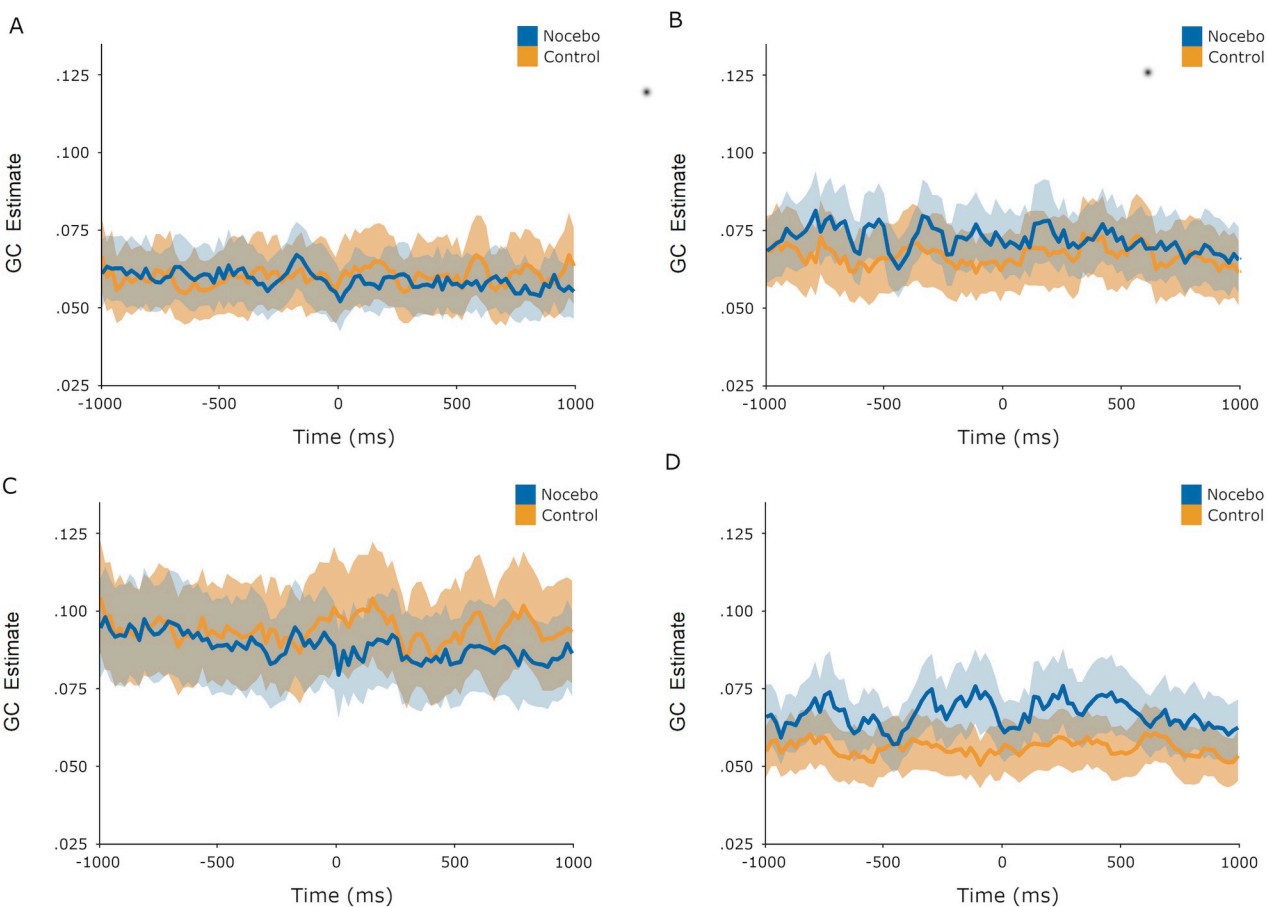

**Fig 4. GC time series during evocation phase.** Evocation phase Granger Causality estimates from -1000ms to +1000ms relative to pain stimulus onset. The shaded area around the plotted lines indicates +/- 1 SEM. A: Frontal left to temporoparietal left. B: Frontal left to temporoparietal right. C: Frontal right to temporoparietal left. D: Frontal right to temporoparietal right.

to shape physical sensations like pain [3, 18]. Given that this one significant result is accompanied by other null findings for the same hypothesis, it should be interpreted cautiously. Future research on this topic could focus on analyzing a time series of GC estimates instead of mean values, as these provide a more granular view of what may be a small effect. Future research could also build on the present study by investigating GC estimates for specific frequency bands instead of taking all frequency bands together. Alpha band oscillations are thought to capture the electrophysiological depiction of pain processing [48], nocebo effects on pain in particular [28], and could be a worthwhile topic of future study.

No relationships between fear of pain and self-reported nocebo effects on pain, SPN, or GC measures were detected. Although there was no apparent link between fear of pain and nocebo hyperalgesia in this study, the sample as a whole was not particularly fearful of pain. The present study was also likely underpowered to detect such relationships, should they exist. In the future, a comparison between low and high fear of pain individuals (e.g., fear of pain scores in the bottom quartile compared to those in the top quartile of responses) in their responses to a nocebo induction paradigm would offer better insights into the potential relationship between fear and nocebo effects. Measuring present-state fear throughout a nocebo conditioning paradigm instead of trait fear of pain may also provide a better understanding of a potential fear-nocebo relationship.

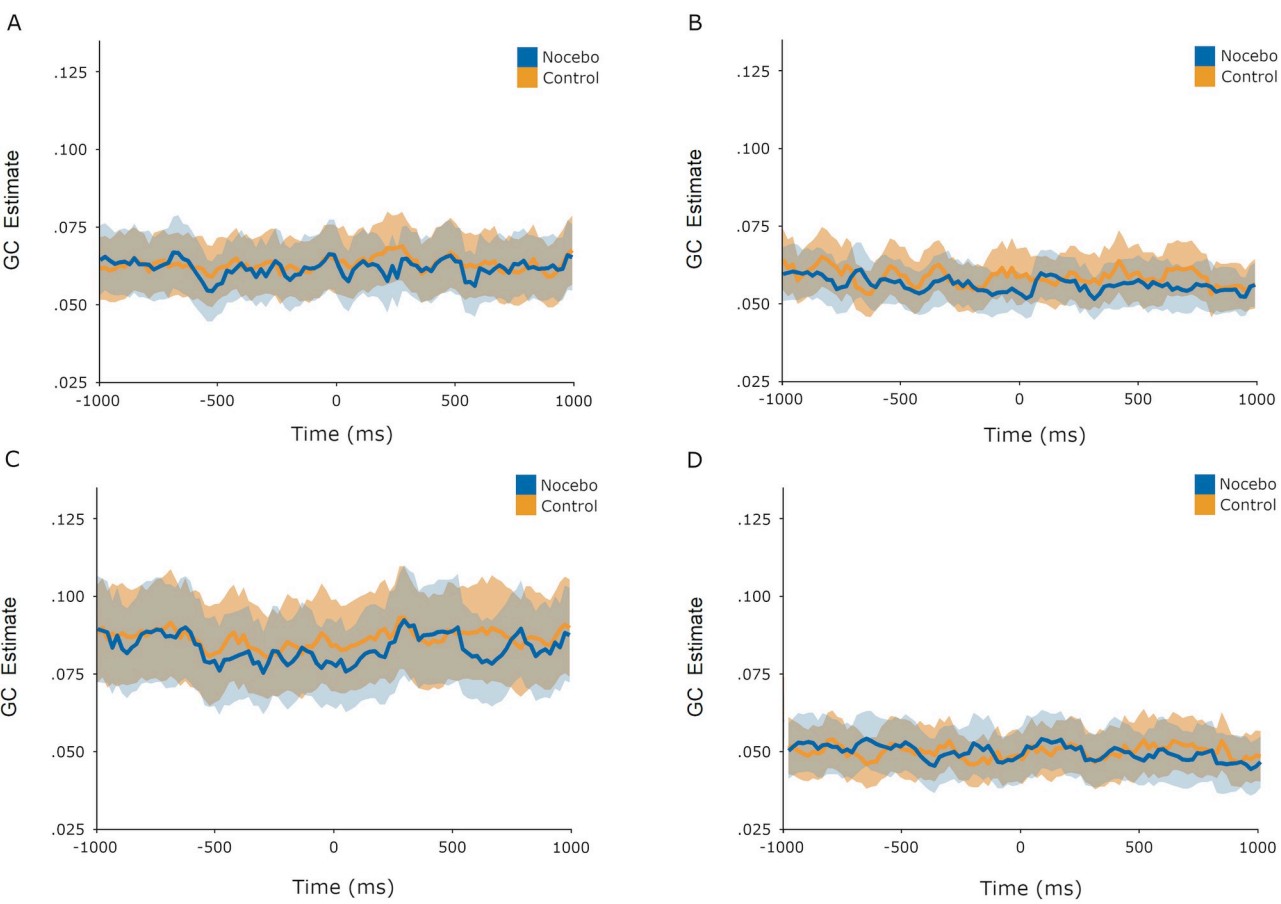

**Fig 5. GC time series during acquisition phase.** Acquisition phase Granger Causality estimates from -1000ms to +1000ms relative to pain stimulus onset. The shaded area around the plotted lines indicates +/- 1 SEM. A: Frontal left to temporoparietal left. B: Frontal left to temporoparietal right. C: Frontal right to temporoparietal left. D: Frontal right to temporoparietal right.

Taken together, these results indicate that while nocebo associated stimuli may coincide with increased anticipatory processing, trait fear of pain does not appear to be linked to this effect. Still, participants may have feared more, or paid greater attention to, the nocebo-associated stimuli relative to control. Whether driven by expectations, fear, attention, or some other cognitive-affective process, both SPN and one of the GC measures indicated that stimuli associated with greater pain could demand greater anticipatory processing relative to less painful stimuli. This pattern of differential anticipatory processing persists even when the actual pain stimulus has been reduced during the evocation phase of the experiment, and may play a role in substantiating the nocebo effect. Future research could further test this proposition by exploring whether fear-associated stimuli, such as images of snakes for snake-phobic individuals, with and without nocebo manipulations, also yield increased SPN amplitudes relative to neutral stimuli, thereby addressing what role fear plays in anticipatory processing for nocebo effects on pain.

Looking beyond the laboratory environment, this study is also relevant for clinical practice. While much attention is rightly paid to the neural profile of pain processing during the actual pain experience, the present study highlights the value of exploring the brain's state prior to pain experiences as well. Though neuroimaging is not likely to be used for detecting nocebo effects in clinical populations, given its expense, the logic of observing an individual's state

prior to their pain symptoms should be extended to clinical settings when considering their propensity to suffer from psychological exacerbation of acute and chronic pain. As has been demonstrated in the fear of pain literature [49], fearful anticipation of pain, and the measures individuals with chronic pain will take to avoid provoking their symptoms can be more disabling than the pain itself [50, 51]. In addition to treating pain itself, this study and the surrounding literature emphasize the importance of studying and treating maladaptive anticipation of pain as well.

In this study, stimulus preceding negativity and Granger Causality were used to measure anticipatory processing and the potential role of expectations for nocebo effects on pain. Both SPN and GC analyses indicated that processing differed between nocebo-augmented pain and control pain without a nocebo component. SPN was more negative, indicating greater anticipatory processing, when expecting nocebo-augmented pain, and there appeared to be increased functional connectivity between the frontal- and temporoparietal-right nodes from one second before to one second after the onset of nocebo pain stimuli. This study provides preliminary evidence that anticipatory processing and top-down activation increases as negative expectations augment sensory processing.

## Acknowledgments

The authors would like to thank Daniëlle Bos for her invaluable assistance to this study.

## Author Contributions

**Conceptualization:** Joseph S. Blythe, Kaya J. Peerdeman, Dieuwke S. Veldhuijzen.

**Data curation:** Joseph S. Blythe.

**Formal analysis:** Joseph S. Blythe, Julian D. Karch.

**Funding acquisition:** Andrea W. M. Evers.

**Investigation:** Joseph S. Blythe.

**Methodology:** Joseph S. Blythe, Kaya J. Peerdeman, Dieuwke S. Veldhuijzen.

**Project administration:** Dieuwke S. Veldhuijzen.

**Software:** Julian D. Karch.

**Supervision:** Kaya J. Peerdeman, Andrea W. M. Evers.

**Validation:** Julian D. Karch.

**Visualization:** Joseph S. Blythe.

**Writing – original draft:** Joseph S. Blythe.

**Writing – review & editing:** Joseph S. Blythe, Kaya J. Peerdeman, Dieuwke S. Veldhuijzen, Julian D. Karch, Andrea W. M. Evers.

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
