## [Decision Letter · Decision Letter 0]

11 May 2023

PONE-D-22-34856Electrophysiological markers for anticipatory processing of nocebo-augmented painPLOS ONE

Dear Dr. Blythe,

Thank you for submitting your manuscript to PLOS ONE. After careful consideration, we feel that it has merit but does not fully meet PLOS ONE’s publication criteria as it currently stands. Therefore, we invite you to submit a revised version of the manuscript that addresses the points raised during the review process. I agree with the reviewer that the manuscript is well done and brings interesting findings of how top-down processing plays into sensory perception. I also agree that the manuscript should be shortened and the English revised by an English-speaking person

We look forward to receiving your revised manuscript.

Kind regards,

Vilfredo De Pascalis

Academic Editor

PLOS ONE

Journal Requirements:

This research was supported by an NWO Vici grant (45316004) and an NWO Stevin award granted to A. Evers.

However, funding information should not appear in the Acknowledgments section or other areas of your manuscript. We will only publish funding information present in the Funding Statement section of the online submission form. 

This work was supported by a Dutch Research Council (NWO) Vici grant (#45316004) awarded to A.W.M. Evers. Website: nwo.nl. The funders had no role in study design, data collection and analysis, decision to publish, or preparation of the manuscript.

Additional Editor Comments:

I agree with the reviewer that the manuscript is well done and brings interesting findings of how top-down processing plays into sensory perception. I agree also agree that the manuscript should be shortened and The English revised by a native English speaker.

Reviewers' comments:

Reviewer's Responses to Questions

**Comments to the Author**

1. Is the manuscript technically sound, and do the data support the conclusions?

Reviewer #1: Yes

2. Has the statistical analysis been performed appropriately and rigorously? 

Reviewer #1: Yes

3. Have the authors made all data underlying the findings in their manuscript fully available?

Reviewer #1: Yes

4. Is the manuscript presented in an intelligible fashion and written in standard English?

Reviewer #1: Yes

5. Review Comments to the Author

Reviewer #1: Overall, the authors do a thorough job describing their study which address a gap in knowledge.

There are some sections that may be trimmed for brevity and some grammatical errors but overall the paper brings interesting findings of how top down processing plays into sensory perception.

6. PLOS authors have the option to publish the peer review history of their article (what does this mean?). If published, this will include your full peer review and any attached files.

Reviewer #1: No

---

## [Author Response · Author response to Decision Letter 0]

9 Jun 2023

Please see the response letter, which provides a detailed and itemized response to all comments from the reviewer, the editor, and the journal.

---

## [Editor Report · Decision Letter 1]

10 Jul 2023

Electrophysiological markers for anticipatory processing of nocebo-augmented pain

PONE-D-22-34856R1

Dear Dr. Blythe,

We’re pleased to inform you that your manuscript has been judged scientifically suitable for publication and will be formally accepted for publication once it meets all outstanding technical requirements.

Kind regards,

Vilfredo De Pascalis

Academic Editor

PLOS ONE

Additional Editor Comments (optional):

I have checked that the authors have addressed the suggested minor changes, improving the study's quality. Thus I think that the manuscript can be accepted for publication.
---

## [Editor Report · Acceptance letter]

18 Jul 2023

PONE-D-22-34856R1 

Electrophysiological markers for anticipatory processing of nocebo-augmented pain 

Dear Dr. Blythe:

I'm pleased to inform you that your manuscript has been deemed suitable for publication in PLOS ONE. Congratulations! Your manuscript is now with our production department. 

Kind regards, 

on behalf of

Prof. Vilfredo De Pascalis 

Academic Editor

PLOS ONE